# Improvement of Body Weight and Nutritional Status in Gastric Cancer Patients Enhances the Benefit of Nivolumab Therapy

**DOI:** 10.3390/jcm11206100

**Published:** 2022-10-17

**Authors:** Tatsuki Ikoma, Toshihiko Matsumoto, Yusuke Kurioka, Masahiro Takatani, Hiroki Nagai, Yusuke Matsumoto, Hironaga Satake, Hisateru Yasui

**Affiliations:** 1Department of Medical Oncology, Kobe City Medical Center General Hospital, Kobe 650-0047, Hyogo, Japan; 2Department of Thoracic Oncology, Kansai Medical University, Hirakata 573-1191, Osaka, Japan; 3Cancer Treatment Center, Kansai Medical University Hospital, Hirakata 573-1191, Osaka, Japan; 4Department of Internal Medicine, Japanese Red Cross Society Himeji Hospital, Himeji 670-8540, Hyogo, Japan; 5Department of Medical Oncology, Kochi Medical School, Kohasu, Nankoku 783-8505, Kochi, Japan; 6Department of Surgery, Japanese Red Cross Society Himeji Hospital, Himeji 670-8540, Hyogo, Japan

**Keywords:** nivolumab, gastric cancer, nutritional status, body weight loss, chemotherapy

## Abstract

Nivolumab improves overall survival (OS) in patients with advanced gastric cancer (AGC) refractory to at least two previous chemotherapy regimens. We investigated whether changes in body weight and nutrition from first-line chemotherapy to nivolumab affected its efficacy. The correlation between weight change and nutritional status up to the start of nivolumab treatment and OS and progression-free survival (PFS) after starting nivolumab treatment was determined. Nutritional status was examined using the C-reactive protein/albumin ratio (CAR). A loss in body weight (LBW) from the onset of the first treatment of <4.5% led to OS prolongation and improved PFS outcomes. The median OS values in the LBW < 4.5% and ≥4.5% groups were 11.4 and 3.6 months, respectively. Similarly, changes in CAR from first-line chemotherapy (ΔCAR) affected OS; the ΔCAR < 0.01 group had a better prognosis than the ΔCAR ≥ 0.01 group. The median OS values in the ΔCAR < 0.01 and ≥0.01 groups were 9.4 and 4.5 months, respectively. The median OS in the group with LBW < 4.5% and ΔCAR < 0.01 was 12.9 months. LBW and deterioration of nutritional status following first-line chemotherapy are poor prognostic factors in AGC patients who received nivolumab as third- or later-line therapy. Early intervention to maintain body weight and nutritional status may improve the efficacy of immune checkpoint inhibitors.

## 1. Introduction

Advanced gastric cancer (AGC) is one of the most common cancers and is particularly common in Asia [1]. In addition, as AGC is often detected when unresectable, the development of corresponding chemotherapy has been of high concern. Recently, the development of immune checkpoint inhibitors (ICIs) has considerably improved the survival of patients with various cancers, including those with AGC [2]. In the ATTRACTION-2 trial, nivolumab improved the overall survival (OS) of patients with AGC that had been refractory to at least two previous chemotherapy regimens [2]. The trial obtained a median OS of 5.26 months. Based on this result, in Japan, nivolumab has become the standard chemotherapy for patients with AGC who had been refractory to at least two previous chemotherapy regimens. Moreover, the advantage of combination chemotherapy and ICIs as first-line chemotherapy has been reported in AGC [3,4]. The ATTRACTION-4 and CheckMate 649 trials demonstrated the efficacy of chemotherapy combined with nivolumab as a first-line setting, showing marked improvements in OS compared with chemotherapy alone [3,4]. Consequently, chemotherapy plus nivolumab has become one of the standard first-line treatments for AGC patients in some countries, including Japan.

In various cancers, including AGC, it has been reported that cancer-induced cachexia, which more often occurs in patients with gastrointestinal cancers who suffer decreased oral intake, reduces the effects of ICIs [5,6,7]. Poor oral intake due to cachexia is associated with weight loss [7]. Furthermore, there have been reports of poor prognosis for chemotherapy in patients with weight loss from the time of presentation to the start of chemotherapy [8]. However, to the best of our knowledge, few studies have investigated the association of weight loss and nutritional status during prior treatment with the effects of ICIs. Therefore, in this retrospective study, we investigated whether loss of body weight (LBW) and worsening nutritional status before the initiation of nivolumab therapy influenced the efficacy of nivolumab in AGC patients.

## 2. Materials and Methods

### 2.1. Patient Characteristics

The clinical data of consecutive patients with AGC treated with nivolumab as the third-line or later treatment were retrospectively collected from the Himeji Red Cross Hospital and Kobe City Medical Center General Hospital. Eligible patients were aged 20 years or older who had advanced, recurrent, or metastatic AGC that was treated with at least one cycle of nivolumab between July 2016 and August 2020. Patient data were evaluated from the date of registration to September 2021. The enrolled patients were administered nivolumab 240 mg every 2 weeks or 480 mg every 4 weeks, until tumor progression or intolerance developed.

### 2.2. Changes in Body Weight and Nutritional Factors

We collected data on body weight changes at three time points: onset of first-line treatment, onset of pretreatment with nivolumab, and onset of nivolumab treatment. Additionally, nutritional status was investigated by evaluating the C-reactive protein/albumin ratio (CAR) [9,10,11] based on blood tests at the three time points. We also investigated the changes in CAR during prior nivolumab treatments.

### 2.3. Ethical Approvals

This study was conducted in accordance with the Helsinki Declaration of 1964 and its later versions and ethical guidelines for clinical studies. This study was approved by the institutional review boards of all participating institutions: Kobe City Medical Center General Hospital (approval no. zh220109) and Himeji Red Cross Hospital. The requirement for informed consent was waived owing to the retrospective nature of the study.

### 2.4. Statistical Analysis

OS was defined as the time from the date of the first nivolumab treatment to the date of death. Living patients were censored at their last follow-up visits. Progression-free survival (PFS) was defined as the time from the date of the first nivolumab treatment to the date of exacerbation confirmed using computed tomography or death for any reason. Computed tomography-based disease assessment was usually performed every 8 weeks, based on the Response Evaluation Criteria in Solid Tumors (version 1.1). Fisher’s exact test was used to compare the patient characteristics. OS and PFS were estimated using the Kaplan–Meier method. The log-rank test was used to compare groups, whereas Cox regression models were used to calculate the hazard ratio (HR) and 95% confidence interval (CI). Patient characteristics and nutritional factors were analyzed using the Cox regression models. OS prolongation was defined as a survival of at least 5.26 months with reference to the ATTRACTION-2 trial [2]. A receiver operating characteristic (ROC) curve was used to determine the correlation between OS prolongation and changes in body weight or nutritional factors as well as the cut-off values according to prolonged OS. Predictive performance was evaluated using the area under the ROC curve (AUC).

Statistical analyses were performed using the SPSS software (version 28.0, SPSS Inc., Chicago, IL, USA), and a *p*-value < 0.05 was considered to be statistically significant.

## 3. Results

### 3.1. ROC Curve Results for Predicting OS by Changes in the Rate of LBW and CAR Value

ROC curves were used to assess the relationship of the rate of LBW from the onset of first-line treatment and onset of nivolumab pretreatment with OS. The AUC values for LBW from the onset of first-line treatment and onset of nivolumab pretreatment were 0.712 and 0.590, respectively. 

In addition, ROC curves were used to assess the impact of the change in the CAR value (ΔCAR) from the onset of the first treatment as well as from the onset of pretreatment to the onset of nivolumab on OS. The AUCs were 0.621 for ΔCAR from the onset of the first treatment and 0.578 from the onset of pretreatment. These results suggest that changes in LBW and CAR from first-line treatment may impact OS. We determined their optimal cut-off values as 4.5 and 0.01, respectively, and used these values to group patients based on LBW and CAR changes.

### 3.2. Patient Characteristics

Between July 2016 and August 2020, 98 consecutive patients with AGC were treated with nivolumab as the third-line or later treatment. The 98 patients were divided into two groups based on the cut-off value of LBW determined by the ROC curve: the LBW < 4.5% group (*n* = 50) and the LBW ≥ 4.5% group (*n* = 47). The characteristics of the patients in each group are summarized in Table 1. There were considerably fewer cases of low BMI and many cases of previous surgery in the LBW < 4.5% group. There were no marked differences in other patient background factors or the efficacy of nivolumab therapy between the two groups. Furthermore, the duration from first-line treatment to the onset of nivolumab treatment was not statistically different in those groups (14.1 months [3.7–103.3] vs. 15.0 months [1.9–53.4], *p* = 0.43).

### 3.3. Correlation of OS with LBW and CAR

With a median observation period of 5.0 (range, 1.8–44.2) months for censored cases, the median OS in the LBW < 4.5% group (11.4 [95% CI 6.6–14.3] months) was significantly greater than that in the LBW ≥ 4.5% group (3.6 [95% CI 2.2–5.1] months) (HR 0.42; 95% CI 0.26–0.66; *p* < 0.001) (Figure 1A). Additionally, the median OS in the ΔCAR < 0.01 group (9.4 [95% CI 5.1–13.7] months) was significantly longer than that in the ΔCAR ≥ 0.01 group (4.5 [95% CI 4.0–5.0] months) (HR 0.59; 95% CI 0.37–0.93; *p* = 0.002) (Figure 1B). These results suggest that minimizing weight loss and maintaining patients’ nutritional status before the initiation of nivolumab therapy may prolong OS. We defined the well-nutrition (WN) group patients as patients who maintained their body weights and remained well-nourished (ΔCAR < 0.01) from the onset of first-line treatment and before the onset of nivolumab therapy. The OS was prolonged significantly in the WN group as compared with the not well-nutrition (NWN) group (median OS, 12.9 vs. 4.5 months; HR 0.43; 95% CI 0.26–0.73; *p* = 0.001) (Figure 1C).

### 3.4. Univariate and Multivariate Analyses of Various Factors Associated with OS

Table 2 lists the univariate and multivariate analyses of prognostic OS factors. Eastern Cooperative Oncology Group Performance Status < 2, LBW < 4.5%, ΔCAR < 0.01, and the absence of peritoneal metastasis may be factors predictive of a better prognosis, while no other factors were prognostic in this study. Among these prognostic factors, LBW < 4.5% correlated strongly with OS.

### 3.5. Correlation of PFS with LBW and Nutritional Status in Univariate and Multivariate Analyses

Similarly, we analyzed PFS in the three groups (Figure 2). In the LBW < 4.5% group, PFS was significantly higher than that in the LBW ≥ 4.5% group (median PFS, 2.7 vs. 2.0 months; HR, 0.63; 95% CI, 0.42–0.94; *p* = 0.021) (Figure 2A). There was also a trend toward improved PFS outcomes in the WN group, although the difference was not statistically significant (HR, 0.63; 95% CI, 0.39–1.01; *p* = 0.050) (Figure 2C). The univariate and multivariate results suggested that LBW < 4.5% might contribute to prolonged PFS (Table 3).

## 4. Discussion

We examined retrospectively the relationship between changes in body weight and nutritional status with the efficacy of nivolumab for AGC. To our knowledge, no previous study has shown that weight loss and worsening nutritional status from the onset of first-line chemotherapy to the onset of nivolumab treatment may affect the efficacy of body weight maintenance. Moreover, we showed that good nutritional status improved OS after initiation of nivolumab. Our study suggests that if LBW can be stopped early and nutritional status can be maintained, OS after the initiation of treatment in AGC could be greatly enhanced. 

LBW has been reported to affect prognosis during other chemotherapy regimens for AGC [12]. However, in our study, we focused on the use of nivolumab and investigated the importance of LBW during treatment prior to nivolumab. Our results showed that LBW < 4.5% after the initiation of first-line therapy was associated with improvements in OS and PFS in AGC patients who subsequently received nivolumab treatment. We then investigated the effect of nutritional status before the initiation of nivolumab treatment. We evaluated the CAR outcome, which is a simple nutritional index that has previously been reported to correlate with prognosis in cancer patients. We found that minimal changes in the CAR from first-line chemotherapy to the onset of nivolumab (ΔCAR < 0.01) correlated with a better prognosis and that improvement or maintenance of nutritional status before the onset of nivolumab had a positive impact on prognosis. Thus, this study suggests that maintaining nutritional status during treatment prior to nivolumab has a positive effect on patients during subsequent nivolumab treatment.

We found that the CAR is a promising indicator of nutritional status. Cancer induces inflammatory cytokines, such as interleukin (IL)-6 and tumor necrosis factor (TNF)-α, and causes LBW [6,9,13]. Therefore, nutritional indicators reflecting these cytokines may indicate cancer cachexia more sensitively. The CAR has been established as a prognostic indicator in patients with acute disease [14] and is a useful prognostic factor in various cancers, including AGC [15]. IL-6 induces C-reactive protein production and affects cancer cell proliferation, invasion, metastasis, angiogenesis, and resistance to treatment via the JAK/STAT3 pathway [13]. In addition, albumin is not only a nutritional indicator, but also an indicator of inflammation in the presence of inflammatory cytokines, such as IL-8. Therefore, the CAR might be an effective marker for investigating both nutrition and inflammation. It has been reported to be a better prognostic indicator for ICI treatment than other inflammatory factors, because it reflects IL-6 levels [16]. 

Based on our results, it is important to maintain weight loss below 4.5% and changes in the CAR value below 0.01. There were no clear cut-off values for the two factors, although our study may suggest one indicator. However, it is particularly difficult to improve the nutritional status of patients with gastrointestinal cancer who have poor oral intake and who consequently develop cachexia [17]. Moreover, in a previous report, the existence of cancer cachexia was associated with a poor clinical outcome after nivolumab treatment in AGC [5]. Anamorelin has been developed as a selective and novel oral ghrelin-like agonist [18]. Ghrelin is an endogenous peptide secreted primarily from the stomach that binds to its receptors and stimulates multiple pathways that regulate body weight, muscle mass, appetite, and metabolism. Anamorelin increases body weight, muscle mass, and appetite in AGC patients with cancer cachexia. While general enteral nutritional supplements and professional nutritional guidance by a dietitian are important, anamorelin may also become an important factor in the future. It should be noted, however, that although the use of anamorelin has shown improvement in nutritional status, there are few data on the contribution of anamorelin itself to survival; therefore, further studies are required. Incidentally, none of the patients in this study had taken anamorelin before the onset of treatment with nivolumab.

In the univariate and multivariate analyses, the LBW < 4.5% and ΔCAR < 0.01 groups had a better prognosis after nivolumab treatment (median OS, 12.9 months). In addition, the LBW < 4.5% group demonstrated better nivolumab efficacy. Our results suggest that OS and PFS in the good nutritional status group tended to be superior to those reported in the ATTRACTION-2 trial. Therefore, our study suggests that maintaining nutritional status may influence the effects of ICIs. 

Our study, however, had some limitations. First, only a few patients were included in our retrospective study. Second, the timing of the computed tomography scans varied from one case to another.

## 5. Conclusions

Our study suggests that maintaining nutritional status during previous treatments may improve the effectiveness of subsequent ICI treatment. Nevertheless, further research is required in this regard.

## 6. Patents

This section is not mandatory but may be added if there are patents resulting from the work reported in this manuscript.

## Figures and Tables

**Figure 1 jcm-11-06100-f001:**
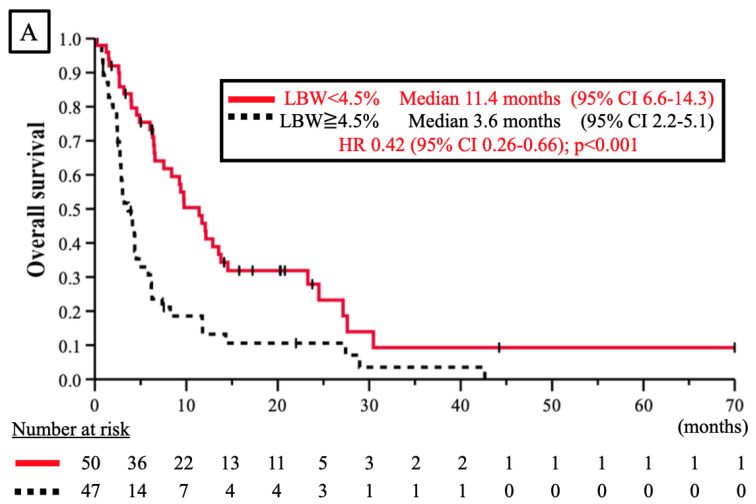
(**A**) Overall survival curve for nivolumab-treated advanced gastric cancer patients classified by the rate of loss in body weight (LBW) from the start of first-line treatment to the start of nivolumab treatment (LBW < 4.5%). Red line: LBW < 4.5% group; black dashed line: LBW ≥ 4.5% group; (**B**) Overall survival curve for nivolumab-treated advanced gastric cancer patients classified by the change in the value of the C-reactive protein/albumin ratio (CAR) from the start of first-line treatment to the start of nivolumab treatment (ΔCAR < 0.01). Blue line: CAR < 0.01 group; black dashed line: CAR ≥ 0.01 group; (**C**) Overall survival curve for nivolumab-treated advanced gastric cancer patients classified by well-nutrition (WN) status. The WN group was defined as patients who maintained their body weight and nutrition from the onset of first-line treatment to the onset of nivolumab treatment (WN vs. non-WN [NWN]). Black line: WN group; black dashed line: NWN group.

**Figure 2 jcm-11-06100-f002:**
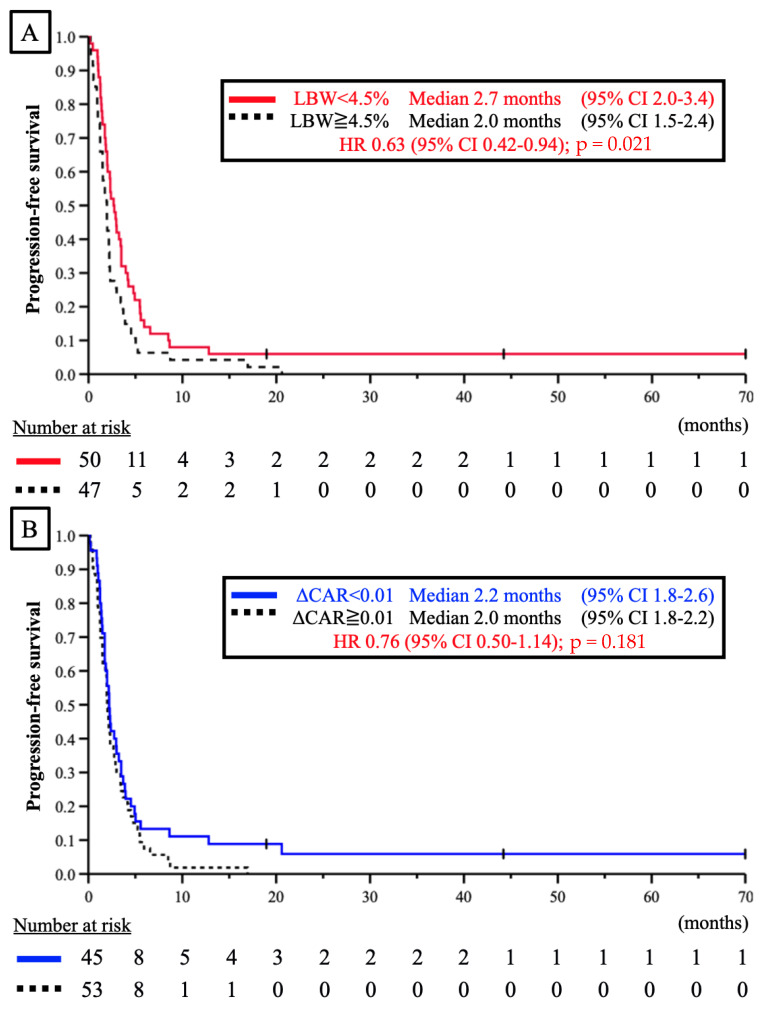
(**A**) Progression-free survival curve for nivolumab-treated advanced gastric cancer patients classified by the rate of LBW from the onset of first-line treatment to the onset of nivolumab treatment (LBW < 4.5%). Red line: LBW < 4.5% group; black dashed line: LBW ≥ 4.5% group; (**B**) Progression-free survival curve for nivolumab-treated advanced gastric cancer patients classified by the change in the value of the CAR from the onset of first-line treatment to the onset of nivolumab treatment (ΔCAR < 0.01). Blue line: CAR < 0.01 group; black dashed line: CAR ≥ 0.01 group; (**C**) Progression-free survival curve for nivolumab-treated advanced gastric cancer patients classified based on the WN status. The WN group was defined as patients who maintained their body weight and nutrition from the onset of first-line treatment to the onset of nivolumab (WN vs. non-WN [NWN]). Black line: WN group; black dashed line: NWN group.

**Table 1 jcm-11-06100-t001:** Patient characteristics.

Characteristic	LBW < 4.5% *n* = 50 (%)	LBW ≥ 4.5% *n* = 47 (%)	*p*-Value
Age (years)	68 [37–85]	70 [31–86]	0.42
Median [range]			
Sex	39/11 (78/22)	38/9 (81/19)	0.81
Male/Female			
ECOG PS	38/12 (76/24)	31/16 (66/34)	0.37
<2/≥2			
BMI (kg/m^2^)	8/42 (16/84)	20/27 (43/57)	0.01
<18.5/≥18.5			
Location	8/21/19/2 (16/42/38/4)	12/23/10/2 (26/49/21/4)	0.30
U/M/L/other			
Histology	27/23 (54/46)	20/27 (57/43)	0.31
Diffuse/Intestinal			
HER2	7/43 (14/86)	11/35 (24/76)	0.30
Positive/Negative			
Previous surgery	29/21 (58/42)	17/30 (36/64)	0.04
Yes/No			
Lung metastasis	10/40 (20/80)	6/41 (13/87)	0.42
Yes/No			
Liver metastasis	18/32 (36/64)	16/31 (34/66)	1.00
Yes/No			
Peritoneal metastasis	30/20 (60/40)	28/19 (60/40)	1.00
Yes/No			
Ascites	27/23 (54/46)	31/16 (66/34)	0.30
Yes/No			
No. of previous regimens	40/10 (80/20)	31/16 (66/34)	0.17
2/≥3			

LBW, loss in body weight; ECOG PS, Eastern Cooperative Oncology Group performance status; BMI, body mass index; U, upper; M, middle; L, lower; HER2, human epidermal growth factor receptor; No., number.

**Table 2 jcm-11-06100-t002:** Cox regression analysis for overall survival.

Variable		Univariate Analysis		Multivariate Analysis	
		HR (95% CI)	*p*-Value	HR (95% CI)	*p*-Value
Age	(<65 years vs. ≥65 years)	0.91 (0.52–1.60)	0.75		
ECOG PS	(<2 vs. ≥2)	0.42 (0.25–0.72)	0.002	0.47 (0.28–0.78)	0.004
Histology	(intestinal vs. diffuse)	0.78 (0.40–1.52)	0.47		
Previous surgery	(yes vs. no)	0.63 (0.33–1.01)	0.15		
Liver metastasis	(no vs. yes)	0.58 (0.33–1.01)	0.05		
Peritoneal metastasis	(no vs. yes)	0.46 (0.25–0.82)	0.01	0.43 (0.26–0.71)	0.001
Ascites	(no vs. yes)	0.68 (0.39–1.12)	0.16		
LBW	(<4.5% vs. ≥4.5%)	0.38 (0.23–0.64)	<0.001	0.37 (0.23–0.60)	<0.001
ΔCAR	(<0.01 vs. ≥0.01)	0.38 (0.23–0.64)	0.04	0.67 (0.35–0.91)	0.02

ECOG PS, Eastern Cooperative Oncology Group performance status; LBW, loss of body weight; CAR, C-reactive protein/albumin ratio; HR, hazard ratio; CI, confidence interval.

**Table 3 jcm-11-06100-t003:** Cox regression analysis for progression-free survival.

Variable		Univariate Analysis		Multivariate Analysis	
		HR (95% CI)	*p*-Value	HR (95% CI)	*p*-Value
Age	(<65 vs. ≥65 years)	0.93 (0.56–1.53)	0.76		
ECOG PS	(<2 vs. ≥2)	0.55 (0.34–0.88)	0.01	0.53 (0.33–0.85)	0.008
Histology	(intestinal vs. diffuse)	0.79 (0.46–1.36)	0.39		
Previous surgery	(yes vs. no)	0.78 (0.45–1.33)	0.35		
Liver metastasis	(no vs. yes)	0.58 (0.35–0.96)	0.03	0.62 (0.40–0.98)	0.039
Peritoneal metastasis	(no vs. yes)	0.66 (0.39–1.12)	0.02	0.59 (0.38–0.91)	0.018
Ascites	(no vs. yes)	0.74 (0.45–1.22)	0.24		
LBW	(<4.5% vs. ≥4.5%)	0.57 (0.37–0.90)	0.02	0.55 (0.36–0.85)	0.006
ΔCAR	(<0.01 vs. ≥0.01)	0.82 (0.53–1.27)	0.37		

ECOG PS, Eastern Cooperative Oncology Group performance status; LBW, loss of body weight; CAR, C-reactive protein/albumin ratio; HR, hazard ratio; CI, confidence interval.

## Data Availability

The datasets generated and/or analyzed during the current study are not publicly available but are available from the corresponding author upon reasonable request.

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
