# Peer review of "Improvement of Body Weight and Nutritional Status in Gastric Cancer Patients Enhances the Benefit of Nivolumab Therapy"

_jcm, 2022, doi:10.3390/jcm11206100_

Round 1

Reviewer 1 Report

This is an interesting paper that describes the relationship between nutritional status, using weight loss and CRP/albumin ratio as surrogates, and OS while on Nivolumab. I have just a couple comments:

1. The title, "Improvement of Body Weight and Nutritional Status in Gastric 2 Cancer Patients Enhances the Efficacy of Nivolumab Therapy" is a little misleading in my opinion, as there is no described intervention to improve body weight in this retrospective study. Furthermore, while there appears to be a relationship to overall survival, it's not totally clear that this is related to improved efficacy with Nivolumab. Weight loss, but not CAR, was associated with improved OS, but even though the p value is significant, the absolute benefit was less than one month for PFS. I would revise the title of the manuscript to not be so overreaching. 

2. Is it possible to include in the baseline patient characteristics the median time from diagnosis to initiation of nivolumab? It would be good to know in general if patients who had LBW <4.5% perhaps had longer duration of treatment on other therapies before switching to nivolumab, or if it was the same in both groups. This may indicate that patients with LBW <4.5% perhaps had more favorable tumor biology compared to patients with LBW >4.5% in general.

3. Along the same lines, I don't believe the relationship between prior surgery and LBW <4.5% was discussed. Does this indicate that patients with LBW <4.5% were more likely to have undergone curative intent surgery, but then recurred? Are patients with LBW > 4.5% more likely to have been metastatic from the time of diagnosis?

4. In the methods, it was noted that patients were on nivolumab until tumor progression or intolerance developed. Can you clarify - for those patients with LBW <4.5%, the median OS was 11 months, and the median PFS was 2.7 months. What did these patients receive for treatment between recurrence and death? Did they stay on nivolumab despite tumor progression? If so, would re-word your methods section. 

Author Response

Reviewer 1:

This is an interesting paper that describes the relationship between nutritional status, using weight loss and CRP/albumin ratio as surrogates, and OS while on Nivolumab. I have just a couple comments:

1.The title, "Improvement of Body Weight and Nutritional Status in Gastric Cancer Patients Enhances the Efficacy of Nivolumab Therapy" is a little misleading in my opinion, as there is no described intervention to improve body weight in this retrospective study. Furthermore, while there appears to be a relationship to overall survival, it's not totally clear that this is related to improved efficacy with Nivolumab. Weight loss, but not CAR, was associated with improved OS, but even though the p value is significant, the absolute benefit was less than one month for PFS. I would revise the title of the manuscript to not be so overreaching. 

Response:We would like to thank you for this kind advice. We have tweaked the title accordingly. We have, additionally, described the following:

P1 L1-2 “Improvement of Body Weight and Nutritional Status in Gastric Cancer Patients Enhances the Benefit of Nivolumab Therapy”

  1. Is it possible to include in the baseline patient characteristics the median time from diagnosis to initiation of nivolumab? It would be good to know in general if patients who had LBW <4.5% perhaps had longer duration of treatment on other therapies before switching to nivolumab, or if it was the same in both groups. This may indicate that patients with LBW <4.5% perhaps had more favorable tumor biology compared to patients with LBW >4.5% in general.

Response:We appreciate your kind advice. We compared the duration from the start of first-line treatment to onset of nivolumab in the two groups. There was no statistical difference between those groups. Thus, we judged that there was not enough evidence to confirm the fact that the LBW <4.5% group had more favorable tumor biology. We have added the following to the manuscript accordingly:

P3 L125-127 “Furthermore, the duration from first-line treatment to the onset of nivolumab treatment was not statistically different in those groups (14.1 months [3.7-103.3] vs. 15.0 months [1.9-53.4], p=0.43).”

  1. Along the same lines, I don't believe the relationship between prior surgery and LBW <4.5% was discussed. Does this indicate that patients with LBW <4.5% were more likely to have undergone curative intent surgery, but then recurred? Are patients with LBW > 4.5% more likely to have been metastatic from the time of diagnosis?

Response:We appreciate your kind advice. This fact did not show that there was a higher recurrence rate in the LBW <4.5%, but only that there were more cases of recurrence in the LBW <4.5% group. Similarly, in our study, there were more unresectable advanced gastric cancer patients, at diagnosis, in the LBW > 4.5%. These results might have been affected by the possibility that there were some patients who had poor oral intake because of transit disturbance or other factors.

  1. In the methods, it was noted that patients were on nivolumab until tumor progression or intolerance developed. Can you clarify - for those patients with LBW <4.5%, the median OS was 11 months, and the median PFS was 2.7 months. What did these patients receive for treatment between recurrence and death? Did they stay on nivolumab despite tumor progression? If so, would re-word your methods section. 

Response:We would like to thank you for this kind advice. There were some cases treated with other chemotherapy regimens after nivolumab. However, none of the patients were continued on nivolumab after confirmation of disease progression.

Reviewer 2 Report

Dear Editor,

Thanks for inviting me to review the manuscript jcm-1908500 entitled "Improvement of Body Weight and Nutritional Status in Gastric Cancer Patients Enhances the Efficacy of Nivolumab Therapy".

I appreciate the authors' work to collect these important data and to write its manuscript. The title is interesting and novel and the manuscript is well organized. This is a very good report. I could not find any mistakes in it. The authors also made the pre-print of their work available in Research Square database, and there is no mismatch between these two versions. Although I think it can be published in the current format, I have only some minor comments.

- Please provide a separate "Ethical considerations" section just before the "statistical analysis" section and move the Helsinki Deceleration, IRB approval, and informed consent explanations to this section.

- The SPSS v.28 is for "SPSS Inc., Chicago, IL", it is not "IBM Corp., 99 Armonk, NY, USA" anymore.

- The manuscript needs minor language editing.

Regards

Author Response

Reviewer 2

Thanks for inviting me to review the manuscript jcm-1908500 entitled "Improvement of Body Weight and Nutritional Status in Gastric Cancer Patients Enhances the Efficacy of Nivolumab Therapy". I appreciate the authors' work to collect these important data and to write its manuscript. The title is interesting and novel and the manuscript is well organized. This is a very good report. I could not find any mistakes in it. The authors also made the pre-print of their work available in Research Square database, and there is no mismatch between these two versions. Although I think it can be published in the current format, I have only some minor comments.

- Please provide a separate "Ethical considerations" section just before the "statistical analysis" section and move the Helsinki Deceleration, IRB approval, and informed consent explanations to this section.

- The SPSS v.28 is for "SPSS Inc., Chicago, IL", it is not "IBM Corp., 99 Armonk, NY, USA" anymore.

- The manuscript needs minor language editing.

Response:We would like to thank you for this kind advice. We have added the follow subsection accordingly:

P2 L80-85 “2.3. Ethical approvals

This study was conducted in accordance with the Helsinki Declaration of 1964 and its later versions and ethical guidelines for clinical studies. This study was approved by the institutional review boards of all participating institutions: Kobe City Medical Center General Hospital (approval no. zh220109) and Himeji Red Cross Hospital. The requirement for informed consent was waived owing to the retrospective nature of the study.”

P3 L101-102 “SPSS Inc., Chicago, IL”

Furthermore, the manuscript has been revised by a native English speaker for tone, language, and clarity. Please refer to the revised manuscript accordingly.

Reviewer 3 Report

Although the issue presented by authors is of great importance, I feel that the submitted manuscript needs to be extensively improved. The Introduction and Discussion require extensive English corrections and more detailed description of the impact of systemic therapy on nutritional status. Materials and methods and Results are fair, however presented data is based on relatively small number of cases. The authors should also point out the limitations of the study. After the initial, major revision, the manuscript can be re-reviewed.

Author Response

Reviewer 3

Although the issue presented by authors is of great importance, I feel that the submitted manuscript needs to be extensively improved. The Introduction and Discussion require extensive English corrections and more detailed description of the impact of systemic therapy on nutritional status. Materials and methods and Results are fair, however presented data is based on relatively small number of cases. The authors should also point out the limitations of the study. After the initial, major revision, the manuscript can be re-reviewed.

Response:We would like to thank you for this valuable advice. The manuscript has been revised in its entirety by a native English speaker for tone, language, and clarity. Please refer to the revised manuscript accordingly.. The limitations, including the small number of cases presented in our report, pointed out by reviewer have been stated (P10 L258-259).

Additionally, we have added the following text to the manuscript accordingly:

P2 L57-58 “Furthermore, there have been reports of poor prognosis for chemotherapy in patients with weight loss from the time of presentation to the start of chemotherapy [8]”

Round 2

Reviewer 3 Report

The authors have addressed the comments;